**Data Availability Statement:** Data cannot be shared publicly because of the possibility of identifying individuals based on the content of the interview transcripts. This restriction is reinforced by the University of Cambridge and additional

# Evaluating clinician acceptability of the prototype CanRisk tool for predicting risk of breast and ovarian cancer: A multi-methods study

Stephanie Archer[1]*, Chantal Babb de Villiers[1], Fiona Scheibl[1], Tim Carver[2], Simon Hartley[3], Andrew Lee[2], Alex P. Cunningham[2], Douglas F. Easton[2], Jennifer G. McIntosh[4], Jon Emery[1,4], Marc Tischkowitz[5], Antonis C. Antoniou[2], Fiona M. Walter[1,4]

1 Primary Care Unit, Department of Public Health and Primary Care, University of Cambridge, United Kingdom, 2 Centre for Cancer Genetic Epidemiology, Department of Public Health and Primary Care, University of Cambridge, United Kingdom, 3 Centre for Computational Biology, University of Birmingham, United Kingdom, 4 Centre for Cancer Research and Department of General Practice, University of Melbourne, Australia, 5 Academic Department of Medical Genetics, University of Cambridge, United Kingdom

* saa71@medschl.cam.ac.uk

## Abstract

### Background

There is a growing focus on the development of multi-factorial cancer risk prediction algorithms alongside tools that operationalise them for clinical use. BOADICEA is a breast and ovarian cancer risk prediction model incorporating genetic and other risk factors. A new user-friendly Web-based tool (CanRisk.org) has been developed to apply BOADICEA. This study aimed to explore the acceptability of the prototype CanRisk tool among two healthcare professional groups to inform further development, evaluation and implementation.

### Method

A multi-methods approach was used. Clinicians from primary care and specialist genetics clinics in England, France and Germany were invited to use the CanRisk prototype with two test cases (either face-to-face with a simulated patient or via a written vignette). Their views about the tool were examined via a semi-structured interview or equivalent open-ended questionnaire. Qualitative data were subjected to thematic analysis and organised around Sekhon's Theoretical Framework of Acceptability.

### Results

Seventy-five clinicians participated, 21 from primary care and 54 from specialist genetics clinics. Participants were from England (n = 37), France (n = 23) and Germany (n = 15). The prototype CanRisk tool was generally acceptable to most participants due to its intuitive design. Primary care clinicians were concerned about the amount of time needed to complete, interpret and communicate risk information. Clinicians from both settings were

approval by the Health Research Authority.
Anonymised transcripts (CanRisk Phase 1) are
available from the Cancer Group at the Primary
Care Unit, University of Cambridge, https://www.
phpc.cam.ac.uk/pcu/research/research-groups/
cancer-group/, for researchers who meet the
criteria for access to confidential data.

**Funding:** This work was supported by Cancer
Research UK grants C12292/A20861; the European
Union's Horizon 2020 research and innovation
programme under grant agreement numbers
633784 (B-CAST) and 634935 (BRIDGES); a
Wellcome Trust Collaborative Award (203477/B/
16/Z); and the PERSPECTIVE programme: The
Government of Canada through Genome Canada
and the Canadian Institutes of Health Research
(grant GPH-129344). Jon Emery is funded by an
Australian National Health Medical Research
Council practitioner fellowship. Marc Tischkowitz
acknowledges funding from the European
Research Council and Cambridge NIHR Biomedical
Research Centre. Fiona Walter is Director of the
multi-institutional CanTest Collaborative which is
funded by Cancer Research UK (C8640/A23385).

**Competing interests:** The BOADICEA model has
been licensed to Cambridge Enterprise for
commercialization, with the authors D.F.E., A.C.A.,
A.P.C., A.L. and T.C. listed as its inventors. These
authors may receive royalties in the future if
commercialization is realized. This does not alter
our adherence to PLOS ONE policies on sharing
data and materials. The other authors declare no
conflicts of interest.

apprehensive about the impact of the CanRisk tool on their consultations and lack of opportunities to interpret risk scores before sharing them with their patients.

## Conclusions

The findings highlight the challenges associated with developing a complex tool for use in different clinical settings; they also helped refine the tool. This prototype may not have been versatile enough for clinical use in both primary care and specialist genetics clinics where the needs of clinicians are different, emphasising the importance of understanding the clinical context when developing cancer risk assessment tools.

## Introduction

In order to maximise opportunities to both prevent and detect cancer early, there is a need to identify people at higher risk, who may benefit from tailored screening and prevention [1–3]. There has been a growing focus on the development of multi-factorial cancer risk prediction algorithms, such as the Breast and Ovarian Analysis of Disease Incidence and Carrier Estimation Algorithm (BOADICEA [4–6]). The most recent version of BOADICEA combines genetic (e.g. rare genetic susceptibility variants and common genetic susceptibility variants in terms of a Polygenic Risk Score), family history, lifestyle and hormonal risk factors to calculate the future risk of developing cancer and the risk of being a carrier of a pathogenic variant [7]. The complex calculations included in BOADICEA have been validated in independent national and international prospective studies demonstrating good discrimination and accuracy [8–10].

Alongside the development of cancer risk prediction algorithms, researchers must also consider the design of graphical user-interfaces (GUIs) that enable the data entry required to populate the algorithm, the presentation of the risks calculated by the algorithm, and guidance on management of prevention strategies for that risk. These are fundamental to the successful implementation of risk prediction models into clinical practice. During the development stages of BOADICEA, a simple web-tool collected information to populate the algorithm. However, as the algorithm became more complex, there was a growing need for a more intuitive interface suitable for use in the clinical setting. As such, a new tool—CanRisk (www. canrisk.org), has recently been developed for healthcare professional use in a variety of clinical settings. Driven by the core principles of user-centred design [11], the core requirements of the CanRisk tool were initially influenced by clinicians from a wide range of clinical settings including primary, secondary and specialist care.

Acceptability is a multi-faceted construct that reflects the extent to which users consider an intervention appropriate, based on anticipated or experienced cognitive and emotional responses [12]. Whilst acceptability is a core facet of user-centred design, it is also an increasingly important component of evaluating the effectiveness of complex interventions [13, 14]. Furthermore, Sekhon et al. [12] suggest that successful implementation depends on the acceptability of the intervention to both intervention deliverers (healthcare professionals in the case of CanRisk) and receivers (patients). Assessing acceptability of any electronic risk assessment tool is particularly important during the beta stage of tool development as it enables further modifications to be made prior to evaluating the clinical utility of risk tools and how best to implement them [3]. Therefore, the aim of this study was to explore the acceptability of the

prototype CanRisk tool among the healthcare professionals for whom it has been designed, in order to optimise further development, evaluation and, ultimately, implementation.

## Methods

### Design(s)

This study used a multi-methods approach. Participants completed a demographics questionnaire, two test cases (either face-to-face with a simulated patient or via a written vignette) and a semi-structured interview or equivalent open-ended questionnaire.

### Participants

General Practitioners and Practice Nurses from primary care and Clinical Geneticists and Genetic Counsellors from specialist genetics clinics in London and the East of England were identified using an initial purposive sampling approach to account for clinical training and region, and enhanced by a snowball approach. Clinicians from specialist genetics clinics in France and Germany who were existing users of the previous simple web-based BOADICEA interface were also invited.

### Ethical considerations

Following the receipt of an invitation to participate and a participant information sheet, participants gave full informed written consent. Participants were given the opportunity to withdraw up to and during study completion. As part of the study debrief, participants were reminded about the process for withdrawing their data. Ethical approval was sought and granted from the Cambridge Psychology Research Ethics Committee to cover all aspects of the study (PRE.2018.011). Additional approval for the study in England was granted by the Health Research Authority (Ref No: 18/HRA/1094); no additional approvals were required for the study in France and Germany.

### CanRisk tool

The CanRisk Tool is a Web-based computer program developed by the Centre for Cancer Genetic Epidemiology at the University of Cambridge, and hosted on a University of Cambridge virtual server. The software is comprised of a number of components that form a series of graphical user interfaces (GUIs) to collect data required to run BOADICEA. This version of the CanRisk Tool was designed for use by healthcare professionals in a range of clinical settings. The tool prompts clinicians to enter the woman's risk factors (e.g. age, lifestyle, hormonal), genetic test results (if known) and family history data, and uses this information to calculate the woman's lifetime risk of breast and ovarian cancer and their risk of being a carrier of a pathogenic variant. Risk calculations are displayed in a variety of formats, including combinations of written text, graphs and icon arrays. For users in the UK, the tool also classifies the woman's risk in accordance with the NICE guidelines [15]. No recommendations for risk management are provided. See Figs 1, 2 and 3 for screenshots of the prototype.

### Procedure

The research design aimed to standardise the content and quality of data collection across countries, so we used two methods of data collection for the study (face-to-face or on-line), determined by proximity to the participants. Participants were identified through local/ national/international research networks (e.g. the Clinical Research Network in the UK) and were initially contacted by email. Participants were eligible for inclusion if they worked in a

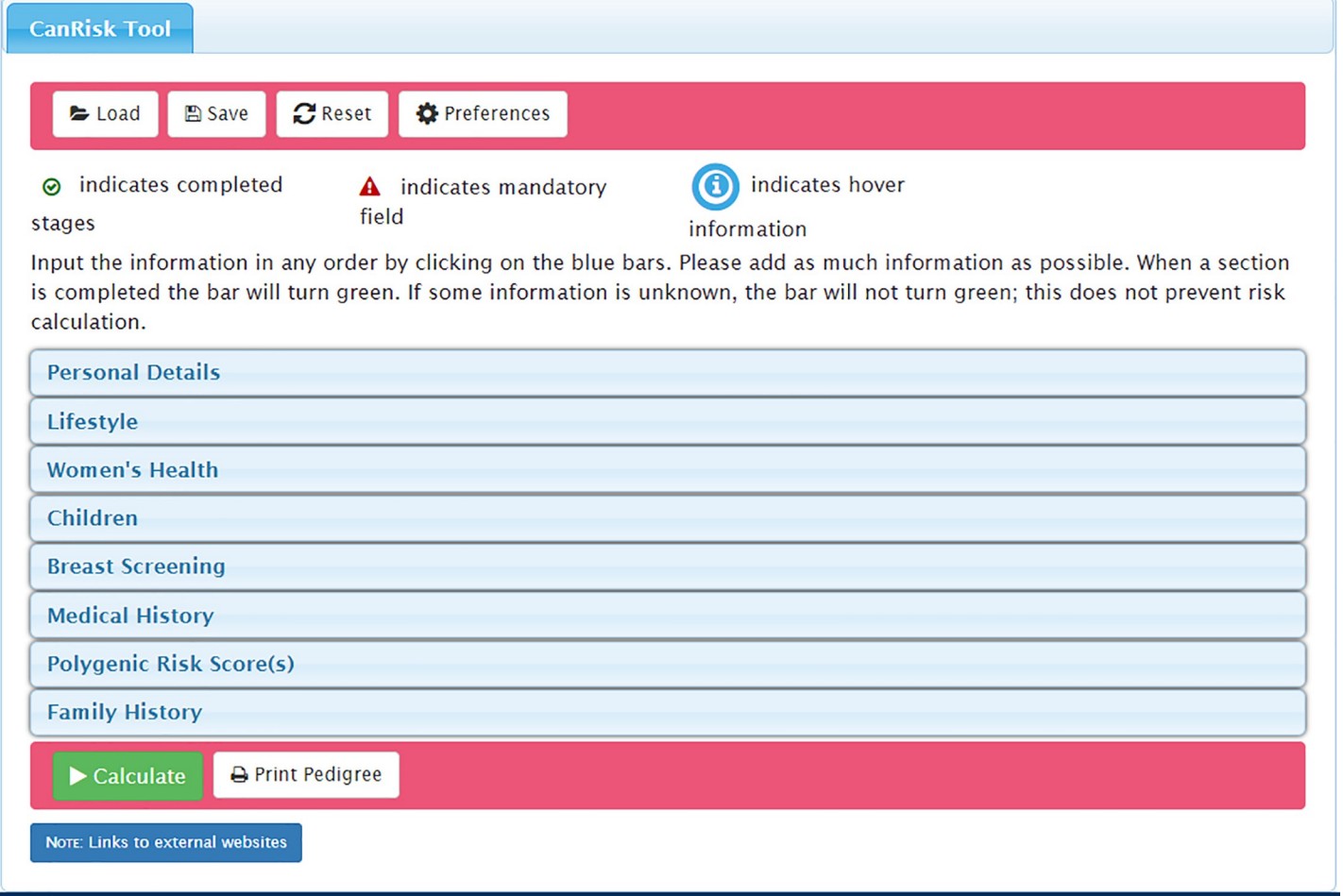

**Fig 1. CanRisk tool initial page displaying the accordion menu of clinical topic areas included in the risk prediction calculation\*.** \* image corresponds to the revised version of the tool (not the beta version assessed).

specialist genetics clinic or general practice and could give informed consent. Data were collected between April 2018 and August 2018, and the study was undertaken in participant's clinical workplaces.

**England.** Those based in England participated face-to-face with the researcher. They completed an initial questionnaire collecting demographic data (age, gender, years in role, number of hours worked, any research qualifications) and information on their current use of computerised risk assessment tools (use of: web browsers, computer in clinic, electronic risk assessment tools, electronic tools to draw family tree). Participants then completed two simulated patient encounters (SPEs) using the CanRisk tool to produce a risk calculation. The SPEs were designed to emulate a typical appointment and were conducted in the participant's normal clinical room with their normal computer. Two female professional medical role players took the part of the patients in the SPEs. Each role player was given a script (see Table 1 for an example) and rehearsed with the research team prior to the study. To standardise the SPEs across multiple participants, the actors were provided with the opening and closing lines of the consultation, information about the patient and their family and a list of comments to include (Table 1). Following completion of the SPE, participants took part in an audio-recorded face-to-face semi-structured interview (the interview schedule is available in S1 File).

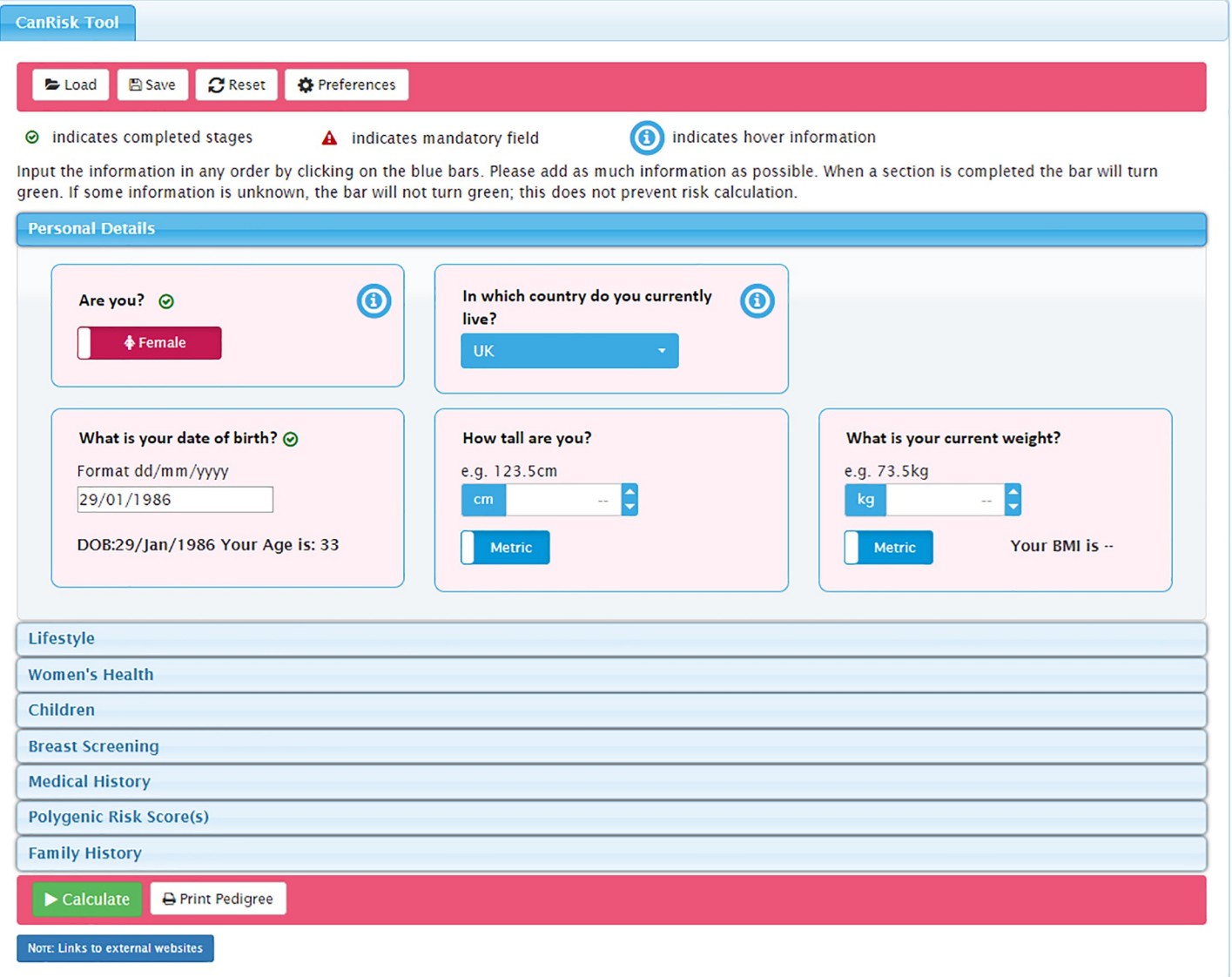

**Fig 2. CanRisk tool detailed view of one topic area: Personal Details\*.** * image corresponds to the revised version of the tool (not the beta version assessed).

**International.** Participants from specialist genetics clinics in France and Germany took part in an online version of the study via the Web-based survey tool Qualtrics [17]. They completed the same initial questionnaire, followed by two vignette-based test cases, adapted from the scenarios used in the English setting but with the same risk factors as in the SPE cases. Demographic information pertaining to a fictitious patient was provided for each case alongside details of their family history (see Table 2 for an example). Participants were asked to enter the information into the CanRisk tool and evaluate how it would work in a real-world setting. Participants were also prompted to take notes as they used the CanRisk tool in order to complete a detailed open-ended questionnaire (again drawn closely from the interview schedule) exploring their user experiences.

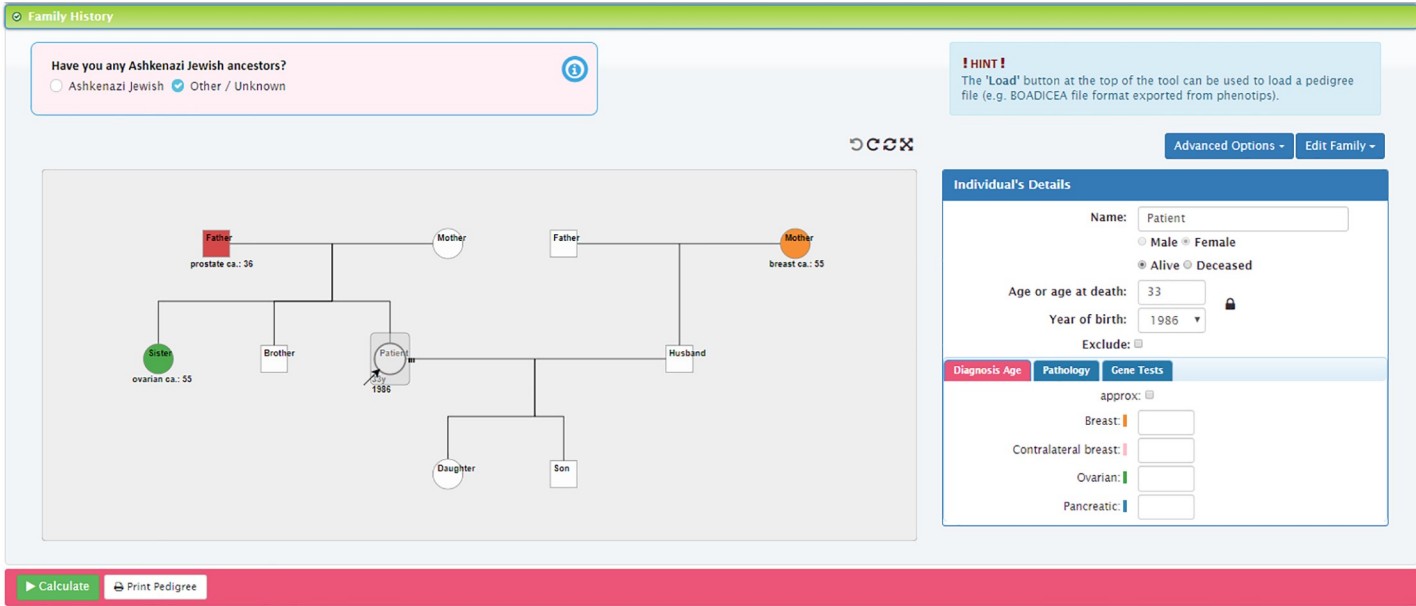

**Fig 3. CanRisk tool detailed view of the interactive graphical pedigree editor [16]\*.** \* image corresponds to the revised version of the tool (not the beta version assessed).

## Analysis

Demographic data and information about current use of computerised risk tools were summarised. Qualitative data from the open-ended questionnaires and verbatim transcriptions from the semi-structured interviews were analysed inductively using thematic analysis [18]. In order to become familiar with the data, two authors (SA and FS) repeatedly read the interview transcripts, and notes were made about potential codes; these were subject to ongoing discussion and refinement. Following this, the same authors performed formal coding using NVivo [19], working through the transcripts and writing notes to support data patterns–these patterns were extracted as codes. Consistency of coding between SA and FS was discussed regularly through team meetings, but formal analysis of the concordance between reviewers was not completed. The codes were drawn together into a broad set of initial themes; these were refined with guidance of the senior author (FW). The final themes were mapped onto Sekhon et al.'s Theoretical Framework of Acceptability [12] which was developed to guide the assessment of acceptability from the perspectives of intervention deliverers and recipients both prospectively and retrospectively. The framework proposed seven component constructs: affective attitude, burden, perceived effectiveness, ethicality, intervention coherence, opportunity costs, and self-efficacy.

## Results

### Demographics

In England, 37 healthcare professionals participated in the study, including 11 General Practitioners (GPs), 10 Practice Nurses (PNs), 6 Clinical Geneticists (CGs) and 10 Genetic Counsellors (GCs) (Table 3). In their patient consultations, most (31/37, 84%) were already using computerised risk tools, relating to cancer (n = 14; GP = 2; CG = 4; GC = 8), cardiovascular (n = 18; GP = 10, PN = 8), and diabetes (n = 4; GP = 1, PN = 3) risk.

**Table 1. Example role player scenario and script used in England.**

| Summary | | Sarah Jones has recently returned to (name and location) after spending a year teaching in Scotland. She has registered with the practice and made an initial appointment to see her GP requesting a mammogram. |
|---|---|---|
| **Personal Details** | Age: | 49 |
| | DOB: | 4/2/1969 |
| | Height: | 5ft 5ins |
| | Weight: | 9.4 stone |
| **Family History** | Daughter: | Olivia (DOB 3/4/97), lives locally. No history of cancer |
| | Mother: | Ann (DOB 2/2/1936). Died 2 years ago, aged 80. Initially diagnosed with breast cancer at 70. Underwent treatment and went into remission. Breast cancer returned aged 80 and she died within a couple of months. |
| | Father: | William aged 82 –No history of cancer |
| | Sister: | Jessica aged 55. She was diagnosed with breast cancer when she was 49. Now in remission. |
| | Maternal Aunt: | Helen aged 78 –no history of cancer |
| | Two paternal Uncles | Aged 78 and 76 –no history of cancer |
| **Medical History** | Children | 1 Pregnancy—Gave birth to Olivia when she was 28. Breast fed for 8 months |
| | Oral contraceptives | Took the oral contraception pill aged 23–25 and 28–30 (6 years total) |
| | First period age | 12 |
| | Last period age | 41 |
| | Current medication | Hormone Replacement Therapy (Climmese)—has been taking for a year) |
| | Alcohol usage | 1 glass of wine every couple of days |
| **Ideas** | | She thinks that she is at high risk of getting breast cancer |
| **Concerns** | | As both her mother and sister have a history of breast cancer, Sarah is worried that she is also at risk. Her sister was diagnosed with cancer when she was 49 and Sarah had her 49th birthday in February. Therefore, it is particularly playing on her mind that she could be at risk of getting cancer too. |
| **Expectations** | | To be given a mammogram. |
| **Opening Line** | | 'I'd like to have a mammogram please.' |
| **Closing Line** | | 'Thank you for seeing me' or 'Thank you for referring me' (depending on the outcome of the discussion) 'What happens next?' |
| **Other comments to include** | | If you see ovarian cancer on the screen–ask about it. 'Why does it say ovarian cancer on the screen?' If shown the graph, point to the upper line and say 'That looks much higher/bigger than expected' |

In France and Germany, 68 people from specialist genetics clinics were approached, and 42 (62%) participated in the study. Four respondents were excluded from the analysis as they were not practising clinicians. The final sample of 38 healthcare professionals comprised 26 Genetic Counsellors (GCs) and 12 Clinical Geneticists (CGs) from specialist genetics services in France (n = 23) and Germany (n = 15). Most (36/38, 95%) were already using computerised risk tools in their consultations to calculate cancer risks.

For the purposes of this analysis, data from GCs and CGs in England, France and Germany were combined into the specialist clinical genetics sample (n = 54). The primary care sample was from GPs and PNs in England alone (n = 21). Detailed participant demographics for each of these groups are displayed in Table 3.

**Table 2. Example vignette based case used in France and Germany.**

| | |
|---|---|
| **Background** | Sarah Jones is a new patient who has just moved from Scotland to London and wants to organise a mammogram because of her family history of breast cancer. |
| **Personal details** | Sarah was born 4 February 1969, is 165cm tall (5'5"), and weighs 60kg (9st4lb, BMI 22.0). She drinks one glass of wine every couple of days. She has no previous diagnosis of breast disease. Her first period was at age 12, and she took an oral contraceptive pill between the ages of 23 and 25, and again aged 28 to 30. She has one child, a daughter (born 1987, when Sarah was 28 years old), who was breastfed for 8 months. She had her last period when aged 41 years, and has taken hormone replacement therapy, Climesse, for the last year. She otherwise takes no medication, has no allergies and is in good health. |
| **Family History** | Sarah's mother (deceased age 80, born 1936) was diagnosed with breast cancer aged 70 and went into remission following treatment; the cancer returned aged 80. Her sister was diagnosed with breast cancer at 49, and she is now a healthy 55 year old. Her relatives are otherwise unaffected, including her daughter, father (aged 82), two paternal uncles (aged 78 and 76), and one maternal aunt (aged 78). |

## Analysis

The thematic analysis generated 17 themes—these were organised around the seven theoretical constructs outlined by Sekhon et al., [12]; Fig 4 shows a thematic map.

**'Affective attitude': How clinicians felt about the CanRisk tool.** Affective attitude was directly influenced by participants' first impressions. For specialist genetics clinicians, the majority of whom had used the previous simple web-based BOADICEA interface, there was a clear improvement:

> *"The CanRisk tool is an improvement to any other risk assessment tool based on BOADICEA. It is straightforward and has a simplistic design, and is therefore intuitive to use"* (GC, Germany).

Clinicians from the specialist genetics setting anticipated that the CanRisk tool would be used frequently before and/or during their clinics, and would influence further testing and treatment decisions. In contrast, whilst some primary care clinicians had a positive outlook on the appropriateness of using the CanRisk tool in their setting, they perceived that the CanRisk tool would not be used frequently:

> *"I thought it was completely something reasonable for a GP to engage in or carry out. So perfectly reasonable consultations and appropriate and relevant. So it's the sort of thing that would happen, not frequently, but I would see these sorts of cases certainly"* (GP, England).

Furthermore, several primary care clinicians were concerned that the CanRisk tool was not contributing to a management plan for their patient, and was simply providing more data on a risk they had already calculated personally:

> *"I'm not sure how useful it is in the primary care setting because all I'm doing is, I'm using that to collect data, and I'm just thinking it's going to calculate my risk and it's not using it to make any clinical decision"* (GP, England).

Participants from both settings expressed views on whether the CanRisk tool could be, and should be, used in face-to-face consultation with patients. Some felt that the CanRisk tool was an asset for structuring complex conversations around factors contributing to cancer risk and involving patients in the consultation:

**Table 3. Participant demographics.**

| | Primary Care | | | Specialist Genetics Clinics | | |
| --- | --- | --- | --- | --- | --- | --- |
| | Total (n = 21) | General Practitioner (GP) n = 11 | Practice Nurse (PN) n = 10 | Total (n = 54) | Clinical Geneticist (CG) n = 18 | Genetic Counsellor (GC) n = 36 |
| **Age** | | | | | | |
| Under 30 | **2** | 2 | 0 | **13** | 1 | 12 |
| 31–40 | **8** | 5 | 3 | **24** | 9 | 15 |
| 41–50 | **5** | 3 | 2 | **13** | 7 | 6 |
| 51+ | **6** | 1 | 5 | **4** | 1 | 3 |
| **Gender** | | | | | | |
| Male | **3** | 3 | 0 | **11** | 5 | 6 |
| Female | **18** | 8 | 10 | **43** | 13 | 30 |
| **Years of Experience** | | | | | | |
| Mean | **11.1** | 9.7 | 12.6 | **7.3** | 7.8 | 6.8 |
| SD | **11.0** | 8.8 | 13.3 | **5.7** | 5.9 | 5.5 |
| Range | **0 to 45** | 1 to 25 | 0 to 45 | **0 to 25** | 2 to 23 | 0 to 25 |
| **Current users of a computerised risk tool** | | | | | | |
| Cancer | **2** | 2 | 0 | **48** | 15 | 33 |
| Cardiovascular | **18** | 10 | 8 | **0** | 0 | 0 |
| Diabetes | **4** | 1 | 3 | **0** | 0 | 0 |
| Other | **8** | 5 | 3 | **0** | 0 | 0 |
| **Nationality** | | | | | | |
| English | **21** | 11 | 10 | **16** | 6 | 10 |
| French | **0** | 0 | 0 | **23** | 3 | 20 |
| German | **0** | 0 | 0 | **15** | 9 | 6 |

*"It's a good way of introducing conversation about risk and about risk perception. And as long as you're doing it with them looking at the screen, so there's that involvement"* (GC, England).

Others were more concerned about using the CanRisk tool alongside patients due to the 'real time' nature of risk communication. Some clinicians described their anxieties about the lack of opportunity to interpret the risk scores before sharing the relevant information with the patient:

*"I wasn't sure what the risk was going to be calculated as, so I didn't know what to expect, and we were going to get that at the same time, and I didn't know how to convey that information when you can both see the numbers there and then"* (GC, England).

In order to provide the best experience of care for patients in this time pressured environment, participants highlighted that they would need to be confident in their understanding, interpretation and communication of multi-factorial cancer risk; this is discussed further in a later theme on self-efficacy.

**'Ethicality': How the CanRisk tool fits with the clinician's value systems.** Ethicality encompassed participants' concerns about how the CanRisk tool changed the dynamic of the consultation. Clinicians from both settings were worried about issues such as reduced eye contact, reduced time spent building rapport, and an altered flow to their normal consultation.

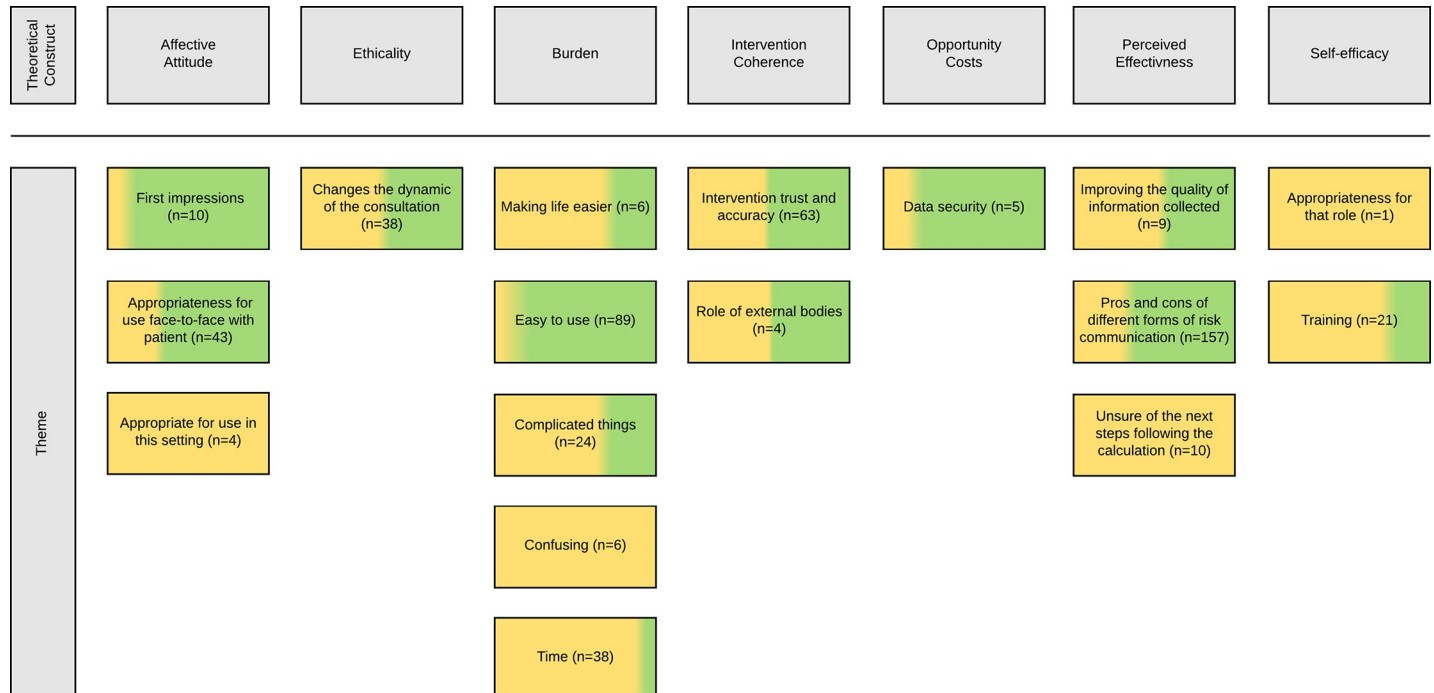

**Fig 4. Inductive themes mapped onto Sekhon et al's Theoretical Framework of Acceptability [12].** The total number of times the theme occurred is presented after each theme name (e.g. n = x). The yellow (primary care) and green (specialist genetics clinic) colouring shows the proportional contribution of each sample to the theme.

*"I think my interaction and communication with the patient was terrible. I didn't get to interact with them, I didn't get to deal with their anxieties around the issue. I was more focusing on inputting data into the tool"* (GP, England).

Further to this, participants from the primary care setting were concerned about the depth and nature of some of the questions, particularly around pregnancy and baby loss, describing them to be intrusive and insensitive:

Participant: *"That's a bit horrible, isn't it? Please include stillbirths, but it's not necessary to include miscarriages".*

Interviewer: *"How could we improve that do you think?"*

Participant: *"I don't know. You want it to be clear and it's better to be upfront. It's easy for me in that I can see what you want, but then it's a little bit. . . to ask that question. I didn't ask either of those women that question"* (PN England).

Clinicians in the specialist genetic setting were also concerned that the patients may feel judged for engaging in particular behaviours (i.e. alcohol), being overweight or for not breast-feeding their child:

*"I'm thinking if I ask questions about breastfeeding and alcohol, it's going to make no difference to the screening recommendations we make. It's actually just going to make someone go away feeling a bit shit because they were told they drink too much and they didn't breastfeed"* (GC, England).

In order to allay these concerns, participants discussed the need for clear explanation about the CanRisk tool, how it works, and why particular types of information were needed in order to achieve the best possible risk estimation. To this end, time in the consultation to achieve this (or a lack of it in many cases) was seen as a major limitation, particularly in primary care, but not a complete barrier to use.

**'Burden': The effort required to use the CanRisk tool.** All participants discussed burden, or the alleviation of it. Some primary care participants described how the CanRisk tool could improve their clinical practice, and that ease of use was an important factor:

> *"It's quite simple to use and I like the fact that it's got the links, so I can link straight over to the NICE guidelines, I can go straight to the public health and look at the breast screening programme"* (PN, England).

Other primary care participants felt that the CanRisk tool was confusing, particularly the aspects that were unfamiliar in their normal clinical practice (e.g. the completion of extended and detailed family histories); in some cases, they were concerned that this could impact on the risk calculation:

> *"It's [family history] quite difficult to put in and then if we're guessing, how accurate is the result? So, potentially, I could be putting someone in a much higher category than they actually are and then they're going to go away and worry, and then they go to the hospital and actually they say, no you're low risk"* (PN, England).

Participants from both settings thought that using the CanRisk tool in clinical practice could complicate the consultation, particularly when patients might not have all of the information readily available. Clinicians from the specialist genetics setting felt that the CanRisk tool was unnecessarily complicated for a simple risk stratification as most of their patients have significant family histories that would not be moderated by other risk factors:

> *"So, this tool, in a way, made things a little more complicated for me than it needed to be, I suppose: there are certain categories of women that automatically will fall into that high risk, for me"* (CG, England).

Primary care clinicians were clearly concerned about the amount of time needed to complete the CanRisk tool. They were apprehensive about their ability to complete the tool, achieve a good patient experience and initiate future management planning within a short consultation:

> *"It's just whether you have time to do it. I don't think you will have time to go through the whole thing without sort of establishing rapport initially, and finishing and closing at the end"* (GP, England).

Specialist genetics clinicians rarely described the time needed to complete the tool as a limiting factor. Indeed, they reported having fewer time pressures due to the nature of their clinical set-up, which generally features longer appointments (up to 45 minutes in England) and often affords greater opportunities to prepare for each consultation.

**'Intervention coherence': Understanding the CanRisk tool and how it works.** Intervention coherence focused predominantly on participants' beliefs about the accuracy of the risk

assessment and the trust they had in the model. For some participants, endorsement of the CanRisk tool by NICE or other trusted authorities instilled greater trust:

> "So, from the cancer risk point of view, obviously, Macmillan or anything like that, or Cancer Research I'd be thinking oh yes, or if it's got NICE approval as well, then I think yes, okay, that's good" (PN, England).

Adoption of the CanRisk tool within a clinical system (i.e. a hospital or GP practice) would also give some participants a feeling of assurance that it was appropriate for use in that clinical setting:

> "If [Clinical IT Specialist] has deemed, and the doctors have deemed, it safe to embed into our templates, then I would trust that that's okay for me to use" (PN, England).

Whilst some participants were happy to rely on the risk calculation given by the CanRisk tool because of its reputation in their clinical community, others expressed concerns about receiving a risk calculation that did not fit with their clinical judgement:

> "I have a figure that I'm expecting, and I find that very handy because if it's wildly different, it alerts me that there could be concerns" (CG, England).

Specialist genetics clinicians found a discrepancy between the CanRisk calculation and their clinical judgement particularly challenging, as many were used to calculating risk based solely on family history information:

> "So, you see, when we calculate risk, we don't take that into account, and we don't take HRT into account, or contraceptive pill into account, at the moment. So, it's very much based on the family history. So, actually, having that be a bit higher, it just makes you think, actually, lifestyle factors may play a part, yes" (GC, England).

Some participants were also concerned about the impact of incomplete information about any of the risk factors, particularly lifestyle and hormonal factors, might have on the risk calculation:

> "A lot of personal information is asked. What about patients [for] who we don't have all these items—the cancer risk is not as precise?" (GC, France).

Many of these concerns were underpinned by a lack of knowledge about how the calculation was made, leaving some participants feeling uncomfortable about the reliability and specificity of the risk calculation:

> "[We'd] say, "This is one model we've got, one tool we've got for estimating your risk and it's coming out saying your risk is about two percent." We'd say that all datasets are incomplete, there may be new data and all sorts of things, but this is the current model. So I think in terms of regulation, it's really important that that uncertainty is somehow captured when it's being used in different clinical situations" (GC, England).

Some participants felt that the calculated risk figure needed to include a caveat to better represent the uncertainty. Participants wanted greater transparency to ensure that healthcare

professionals and their patients were fully informed about the limits of both the BOADICEA model and the CanRisk tool.

**'Opportunity costs': The sacrifice of benefits, profits and values.** Opportunity costs were seldom discussed by participants in this study. Only data security, in terms of the safety of data entered into the CanRisk tool, was seen a potential problem. For some, there were concerns about data protection, specifically around that integration of risk prediction tools into existing IT systems:

> *"I think that's a legitimate concern, we have electronic records and some people are worried about that, and so having online tools where you're putting in all of this personal information, putting in your family members, that might spark some patient concerns"* (GC, England).

Other concerns focused on the security of the Web more generally, the chances for data to be hacked, and whether any secondary use of the data was planned. None of the participants reported that these concerns would stop them from using the CanRisk tool, but they acknowledged that these points should be addressed in any guidance documentation.

**'Perceived effectiveness': The ability of the CanRisk tool to achieve its purpose.** Perceived effectiveness was a central tenet of acceptability for most participants in this study, with views on efficacy being linked to positive attitudes towards the CanRisk tool. Some participants believed that the use of the CanRisk tool promoted more thorough data collection and therefore improved the quality of information:

> *"So that's good, because you don't miss information and it's quite tidy. And, in your mind, you can understand where you're going with the information. And, actually, it's not even very alarming to the patient, because they can see where you're headed with the questions"* (GP, England).

Following completion of the risk calculation, many participants talked about the benefits of having varied and versatile formats to help them communicate the patient's risk:

> *"It was really helpful offering various versions [of risk communication]. At least one should fit for all of the patients—it is sometime helpful explaining the same message using different paths"* (CG, Germany).

However, several PNs and GCs felt that the range of options for communicating risk were overwhelming. In some cases, the additional information was felt to be unnecessary as the risk calculation forms just one part of the wider consultation:

> *"Sometimes you won't need to go into any more detail than moderately increased risk. Sometimes that's absolutely fine. Moderately increased risk, you know, and actually the rest of the consultation might be about their feelings surrounding coming up to the age at which their mum was diagnosed and their thoughts about that"* (GC, England).

In general, clinicians from the specialist genetics setting were confident about the next steps following a risk calculation; in some cases the tool was used to support a decision for a particular type of treatment (e.g. risk-reducing mastectomy). Conversely, primary care participants talked about their uncertainty about the next steps following the risk calculation and there was a general frustration about the lack of information on clinical management:

*"I would know the risk stratification off the top of my head and I would know what to do next with it. But I'd know there is some guidelines somewhere that I need to follow in terms of whether or not they need a referral or not. But if we are going to use a risk stratification tool, then I would want some sort of, and you do this next, it's this is the risk, deal with it, GP"* (GP, England).

Following this, some PNs were concerned about how and when to pass the case over to a GP colleague, particularly if the patient was identified as being at moderate or high risk. Some PNs were conscious of their lack of knowledge of the next steps, and this affected their confidence in being able to provide appropriate information to their patient:

*"I don't even know what, how quickly, people would get assessed for BRCA, I wouldn't know if I send them here, if they'd have to go to Place 1, I really don't know, so, because it's not something I've done"* (PN, England).

Whilst some participants believed that the CanRisk tool could help regulate the collection of data and structure conversations around risk, for the majority, the desire to develop confidence and competence when using the CanRisk tool was integral to its acceptability.

**'Self-efficacy': Clinician's confidence in using the CanRisk tool.** Self-efficacy was predicated upon a need for additional training. Participants felt that training should focus on how to best use the CanRisk tool, promoting a sense of proficiency:

*"We should be familiar with the tools that we're using, so that we use them properly. And therefore, I think there should be a training component to it. Otherwise, you make mistakes"* (CG, England).

All participants wanted to be able to feel comfortable about using the CanRisk tool in a consultation. Those from the specialist genetics clinics felt more confident due to their experience of using the simple web-based BOADICEA interface. In addition, their specialist training left them comfortable communicating cancer risk to their patients, but some struggled to describe the impact of non-genetic factors. Conversely, whilst most primary care participants were confident in their communication skills around multi-factorial risk, some were concerned about their limited genetics knowledge:

*"Genetics, not part of my normal knowledge. Even in medical school you do a tiny, weeny bit of it, it's not, that is an absolute specialist area"* (GP, England).

Whilst only one PN said they were '*not so sure this would be an appropriate thing for somebody of my level of nursing to do*' (PN, England), many other primary care participants stated that they would need additional training on interpreting genetic risk and its communication:

*"So you've got to have an education package that goes with it, to tell people what it's about, why all the things are on there and actually how to interpret it. I think most of us, as doctors, my sort of age, never had any consultations since training, we'd have never been taught how to convey risk"* (GP, England).

Whilst the training needs of participants varied depending of the setting and/or the clinical training and experience of the participant, there was commonality in terms of a desire to provide the best care for their patients.

## Discussion

Our findings suggest that the prototype CanRisk tool was broadly acceptable to the majority of participants. Most clinicians found the CanRisk tool easy to use due to its intuitive design. For geneticists and genetic counsellors who were users of the simple web-based BOADICEA interface, CanRisk offered a more user-friendly experience and the standardised format for data collection was of benefit to many. The foremost concern for clinicians from the primary care setting was the amount of time needed to complete and interpret a risk calculation, highlighting a potential barrier to the use of the CanRisk tool in this setting. Clinicians from both settings were also apprehensive about the impact that the CanRisk tool had on their consultation and the lack of opportunity to interpret the risk score before sharing it with the patient.

Many participants found the CanRisk tool to be intuitive; however, the majority were from specialist genetics clinics with previous experience of the simple web-based BOADICEA interface. All participants were concerned about potentially unhelpful changes to their consultation style influenced by the inclusion of detailed questions in the prototype that may appear to be unrelated to cancer risk, and increased time spent looking at the computer screen, consistent with the 'third party effect' [20]. In order to allay these concerns, a period of familiarisation and training prior to using the CanRisk tool for the first time in the clinical setting is advisable. This would allow healthcare professionals to navigate the CanRisk tool confidently, select which risk factors are important for that patient, and collect information on these in a way that is consistent with their consultation style [20, 21].

Many participants were also concerned about collecting the extra information needed for this multi-factorial cancer risk prediction tool. This was particularly evident when these topics are not routinely explored such as lifestyle factors by clinical geneticists and genetic counsellors and extended detailed family histories for GPs and practice nurses. Primary care clinicians were particularly concerned about the amount of information required for risk prediction as they had limited time available during their consultations [22]. In order to improve the acceptability of the CanRisk tool, the burden of data collection must be addressed. Some areas of the CanRisk tool could be pre-populated from existing clinical records; this would require the CanRisk tool to be embedded in electronic medical records (e.g. EMIS or SystmOne for primary care, or EPIC for hospital care). Another approach could be for patients to complete an electronic pro-forma. Whilst these two potential solutions may sound simple, the technical and regulatory requirements may make it difficult to navigate on a local and national level [23, 24], and may add further concerns around data security.

The study findings suggest that endorsement from appropriate clinical bodies is essential for confidence in adopting new technologies into the clinical setting. Whilst BOADICEA is referenced in the NICE clinical guidance for familial breast cancer (CG164) [15], clinicians may not be aware of the association between BOADICEA and the CanRisk tool. Including a reference or link to the CanRisk tool specifically within the guideline may increase visibility of the tool, strengthen the association between BOADICEA and the CanRisk tool, and increase adoption. The addition of CE marking to the CanRisk tool, indicating conformity with health, safety and environment protection standards for products within the European Economic Area [25], is also likely to support acceptability across the clinical settings.

Although some clinicians regarded the variety of risk presentation formats as a useful feature, others found them to be overwhelming. Further consideration of the risk presentation formats for each job role and/or clinical setting may bolster the clinician's confidence, assist in quick interpretation and improve patients' understanding and perception of risk [26]. In order to achieve shared understanding, incorporating layers of information, where details about the risk and its parameters are displayed in an incremental and controlled fashion, may be of

benefit to both patients and staff [27]. Additional learning resources focusing on the communication of multifactorial risk for those in the specialist genetics setting and genetic risk for those in the primary care setting may also support clinicians to communicate the results more effectively. The development and testing of these resources should be considered prior to implementation.

The analysis highlighted a number of factors influencing participants' views on the perceived effectiveness of the final CanRisk tool. Specifically, there were concerns about the validity of the output when there is missing information; more detailed information regarding the complex calculations performed in BOADICEA and the contribution of each risk factor to the model may be required. Information on the amount of error within each calculation should be included to better inform users about the level of uncertainty. This may help develop patient centred understanding of the risk score and cancer risk more broadly and what role this information may have in future care decisions [28]. The analysis also emphasised a need for the CanRisk tool to include clinical guidance on managing patients following a risk calculation. This was particularly important in the primary care setting, where most management of women found to be at population and moderate risk for breast cancer is likely to take place.

## Limitations

This study collected data on acceptability from a variety of clinician groups, using a range of methods; whilst this allowed us to achieve both breadth and depth, it also resulted in a number of limitations. Firstly, this analysis was generated using data from both semi-structured interviews and questionnaires; the use of questionnaires to collect detailed experiential information was pragmatic, but the opportunity to explore some issues in more detail was limited. Secondly, no GPs contributed data in the international sample; this restricted the feedback from an integral group of clinicians and limits the generalisability of this research to the international primary care community. Thirdly, as the majority of clinicians from the specialist genetics clinics had previous experience of using BOADICEA in their clinical practice, some aspects of the CanRisk tool were already familiar; this may have amplified the differences between clinicians from primary care and specialist genetics clinics.

## Next steps

Further development and testing of the CanRisk tool is required. The findings presented here have already underpinned a number of changes to the prototype CanRisk tool (see Table 4). Whilst these changes may go some way to improving the tool for clinical users in a variety of settings, further research exploring the patient experience of the tool and its use in clinical practice is necessary. Data collected in these studies would allow us to understand the acceptability of the CanRisk tool in this important stakeholder group and may inform future patient centred design.

## Conclusion

The prototype CanRisk tool was generally acceptable to our participating clinicians. However, the findings highlight the challenges associated with developing a complex tool for use in different clinical settings. The next version of the CanRisk tool is likely to benefit from having core and clinical setting-specific components to increase acceptability across the clinical settings and, ultimately, future implementation.

**Table 4. Summary of changes made to the prototype CanRisk tool.**

| Area of concern | Setting | Changes made to later version |
|---|---|---|
| Differences in practice between the two settings | Both | The tool has now been split into a CanRisk Genetics version for use in the specialist genetics setting and a CanRisk Core version for use in primary care. |
| Lack of approval from regulatory body | Both | The final version of the tool will carry CE marking. |
| Lack of information on how to manage the risk | Primary care | CanRisk Core will include better signposting to relevant management options and provide links to relevant parts of the NICE guidance. |
| Difficult questions leading to discomfort (e.g. pregnancy and baby loss) or judgement (e.g. breastfeeding, alcohol intake and weight) | Both | Questions collecting data on factors that do not currently contribute to the risk model have been removed.<br>A description of why other data are collected has been included in the FAQs. |
| Completing a detailed family history | Primary care | Collection of a limited family history (i.e. first/second degree relatives) will be required for CanRisk Core. |
| Lack of information about how the data required for the calculation were to be used and stored | Both | Enhanced information on the privacy policy has been included at registration. An additional embedded webpage called *Privacy and Cookies* has also been included with detailed information. |
| Need for training to use the tool effectively | Both | Following registration, users are taken to a quick start guide to familiarise themselves with the main functionality of the tool. In addition, an updated and searchable section of *Frequently Asked Questions* has been included. |
| Further training on genetic risk communication | Primary Care | The CanRisk Core version of the tool will include refined risk communication formats, enabling easier interpretation in the time-limited environment. |

## Supporting information

**S1 File.**
(DOCX)

## Acknowledgments

We would like to thank Maureen Rae, Rita Schmutzler, Jon Roberts, Antoine de Pauw and Hugh Wilson for their contributions to aspects of completing this study.

## Author Contributions

**Conceptualization:** Chantal Babb de Villiers, Douglas F. Easton, Jennifer G. McIntosh, Jon Emery, Marc Tischkowitz, Antonis C. Antoniou, Fiona M. Walter.

**Data curation:** Stephanie Archer, Fiona M. Walter.

**Formal analysis:** Stephanie Archer, Fiona Scheibl, Fiona M. Walter.

**Funding acquisition:** Marc Tischkowitz, Antonis C. Antoniou, Fiona M. Walter.

**Investigation:** Chantal Babb de Villiers.

**Methodology:** Stephanie Archer, Chantal Babb de Villiers, Jennifer G. McIntosh, Jon Emery, Marc Tischkowitz, Antonis C. Antoniou, Fiona M. Walter.

**Project administration:** Stephanie Archer, Chantal Babb de Villiers, Fiona M. Walter.

**Software:** Tim Carver, Simon Hartley, Andrew Lee, Alex P. Cunningham, Douglas F. Easton, Antonis C. Antoniou.

**Supervision:** Douglas F. Easton, Jon Emery, Marc Tischkowitz, Antonis C. Antoniou, Fiona M. Walter.

**Writing – original draft:** Stephanie Archer, Fiona M. Walter.

**Writing – review & editing:** Stephanie Archer, Chantal Babb de Villiers, Tim Carver, Simon Hartley, Andrew Lee, Alex P. Cunningham, Douglas F. Easton, Jennifer G. McIntosh, Jon Emery, Marc Tischkowitz, Antonis C. Antoniou, Fiona M. Walter.

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
