## [Decision Letter · Decision Letter 0]

2 Jan 2020

PONE-D-19-33335

Evaluating clinician acceptability of the prototype CanRisk tool for predicting risk of breast and ovarian cancer: a multi-methods study

PLOS ONE

Dear Drs. Archer,

Thank you for submitting your manuscript to PLOS ONE. After careful consideration, we feel that it has merit but does not fully meet PLOS ONE’s publication criteria as it currently stands. Therefore, we invite you to submit a revised version of the manuscript that addresses the points raised during the review process.

We would appreciate receiving your revised manuscript by Feb 16 2020 11:59PM. To enhance the reproducibility of your results, we recommend that if applicable you deposit your laboratory protocols in protocols.io, where a protocol can be assigned its own identifier (DOI) such that it can be cited independently in the future. For instructions see: http://journals.plos.org/plosone/s/submission-guidelines#loc-laboratory-protocols

We look forward to receiving your revised manuscript.

Kind regards,

Alvaro Galli

Academic Editor

PLOS ONE

Journal Requirements:

3. Please provide additional details regarding participant consent.

In the ethics statement in the Methods and online submission information, please ensure that you have specified what type of consent you obtained (for instance, written or verbal, and if verbal, how it was documented and witnessed). If your study included minors, state whether you obtained consent from parents or guardians.

4. Thank you for sending us the data set underlying the results presented in your PLOS ONE submission. We notice that some of the information included in the data set may be potentially identifying. Please ensure that the data shared are in accordance with participant consent and provide only the data that are used in this specific study. To ensure patient confidentiality, we would recommend removing any identifying information in Box 2 of the manuscript. Additional guidance on preparing raw clinical data for publication can be found in our Data Policy FAQs (https://journals.plos.org/plosone/s/data-availability#loc-clinical-data).

5. In your Methods section, please provide additional information about the participant recruitment method and the demographic details of your participants. Please ensure you have provided sufficient details to replicate the analyses such as:

a) the recruitment date range (month and year),

b) a description of any inclusion/exclusion criteria that were applied to participant recruitment,

c) a description of how participants were recruited, and

d) descriptions of where participants were recruited and where the research took place.

6. We note that you have indicated that data from this study are available upon request. PLOS only allows data to be available upon request if there are legal or ethical restrictions on sharing data publicly. For information on unacceptable data access restrictions, please see http://journals.plos.org/plosone/s/data-availability#loc-unacceptable-data-access-restrictions.

7. We note that you have a patent relating to material pertinent to this article.

Please provide an amended statement of Competing Interests to declare this patent (with details including name and number), along with any other relevant declarations relating to employment, consultancy, patents, products in development or modified products etc. Please confirm that this does not alter your adherence to all PLOS ONE policies on sharing data and materials, as detailed online in our guide for authors http://journals.plos.org/plosone/s/competing-interests by including the following statement: "This does not alter our adherence to  PLOS ONE policies on sharing data and materials.” If there are restrictions on sharing of data and/or materials, please state these. Please note that we cannot proceed with consideration of your article until this information has been declared.

8. PLOS requires an ORCID iD for the corresponding author in Editorial Manager on papers submitted after December 6th, 2016. Please ensure that you have an ORCID iD and that it is validated in Editorial Manager. To do this, go to ‘Update my Information’ (in the upper left-hand corner of the main menu), and click on the Fetch/Validate link next to the ORCID field. This will take you to the ORCID site and allow you to create a new iD or authenticate a pre-existing iD in Editorial Manager. Please see the following video for instructions on linking an ORCID iD to your Editorial Manager account: https://www.youtube.com/watch?v=_xcclfuvtxQ

9. Please include captions for your Supporting Information files at the end of your manuscript, and update any in-text citations to match accordingly. Please see our Supporting Information guidelines for more information: http://journals.plos.org/plosone/s/supporting-information

Reviewers' comments:

Reviewer's Responses to Questions

**Comments to the Author**

1. Is the manuscript technically sound, and do the data support the conclusions?

Reviewer #1: Yes

Reviewer #2: Yes

2. Has the statistical analysis been performed appropriately and rigorously? 

Reviewer #1: N/A

Reviewer #2: N/A

3. Have the authors made all data underlying the findings in their manuscript fully available?

Reviewer #1: Yes

Reviewer #2: No

4. Is the manuscript presented in an intelligible fashion and written in standard English?

Reviewer #1: Yes

Reviewer #2: Yes

5. Review Comments to the Author

Reviewer #1: The authors presented the qualitative findings from their study of CanRisk, a web-based version of the BOADICEA model for predicting breast and ovarian cancer risk, among general practitioners and genetic specialists in the UK, France, and Germany. A strength of their study is their use of qualitative data from practitioners to propose changes to their tool to allow for greater applicability in the primary care setting. The authors should address the following comments:

1) The authors should report a kappa on the concordance of the coding of the transcripts between the two coders.

2) In the discussion, the authors described barriers to the use of the tool in the primary care setting largely because of competing demands and time constraints in clinical practice. The authors should acknowledge the limitations of using this tool in a busy primary care practice.

2) In future studies, the authors could consider evaluating the patient experience in using this tool, as patient participation is also vital to the success of such an intervention. There were concerns raised by practitioners using the tool regarding insensitive and intrusive questions (for example, regarding stillbirths and miscarriages); patient feedback in this area would be particularly relevant. The authors also discuss potentially using family history intake via the patient portal and patient feedback would also be helpful in creating this.

3) Another potential challenge to the implementation of this tool raised by authors is the interpretation and communication of risk estimates to patients by primary providers (not the genetics specialists). This is clearly an unmet need in primary care, and education regarding communicating cancer risk and options for risk management are clearly required for this tool to be useful in a generalist practice.

Reviewer #2: This manuscript describes a useful qualitative evaluation of a web-based tool to apply the BOADICEA risk prediction tool in clinical practice. The investigators tested the tool with and received comments from both general practitioners and specialists in genetics clinics, allowing them to assess the strengths and weaknesses of the tool in different practice settings. Overall, the paper was clearly written and the findings were thoughtfully interpreted.

Specific comments on the paper are:

1. Abstract: The summary of the results in the abstract is quite vague and doesn’t highlight some of the key findings described in the paper. In particular, the authors simply name the key themes and don’t really describe some of the major findings including concerns about the time needed to use the tool in primary care or how to address the next steps after risk assessment.

2. Introduction, first sentence (minor comment): The sentence states that “…there is a need to identify people at higher risk for tailored screening and prevention”, which is an ambiguous sentence structure. It would be preferable to state it as “…there is a need to identify people at higher risk, who may benefit from tailored screening and prevention”.

3. Box 1: There is inconsistency in the information presented regarding the subject’s birthdate (4/2/1969), her daughter’s birthdate (3/4/1997) and the age at which she gave birth (18). Based on the birthdates, the mother was 28, not 18, when she gave birth to her daughter.

4. Page 9, Analysis (minor comment): Clarify that “these were refined with guidance of FW” is another author.

5. Results: Although it is recognized that this was a qualitative analysis, it would be useful to quantify how frequently some of the comments were made. For example, how many times did clinicians express concerns about the time it would take to use the tool or how many had concerns about the questions being too intrusive or upsetting. It would help the reader to know whether the concerns were reported by the many clinicians or only by one or two people. If stratified by type of practitioner, it could quickly highlight differences in perceptions and needs by clinical setting.

6. PLOS authors have the option to publish the peer review history of their article (what does this mean?). If published, this will include your full peer review and any attached files.

Reviewer #1: No

Reviewer #2: No

---

## [Author Response · Author response to Decision Letter 0]

5 Feb 2020

We have uploaded a response to reviewer comments document separately.

---

## [Decision Letter · Decision Letter 1]

20 Feb 2020

Evaluating clinician acceptability of the prototype CanRisk tool for predicting risk of breast and ovarian cancer: a multi-methods study

PONE-D-19-33335R1

Dear Dr. Archer,

We are pleased to inform you that your manuscript has been judged scientifically suitable for publication and will be formally accepted for publication once it complies with all outstanding technical requirements.

With kind regards,

Alvaro Galli

Academic Editor

PLOS ONE

Additional Editor Comments (optional):

Reviewers' comments:

Reviewer's Responses to Questions

**Comments to the Author**

1. If the authors have adequately addressed your comments raised in a previous round of review and you feel that this manuscript is now acceptable for publication, you may indicate that here to bypass the “Comments to the Author” section, enter your conflict of interest statement in the “Confidential to Editor” section, and submit your "Accept" recommendation.

Reviewer #1: All comments have been addressed

Reviewer #2: All comments have been addressed

2. Is the manuscript technically sound, and do the data support the conclusions?

Reviewer #1: Yes

Reviewer #2: Yes

3. Has the statistical analysis been performed appropriately and rigorously? 

Reviewer #1: N/A

Reviewer #2: N/A

4. Have the authors made all data underlying the findings in their manuscript fully available?

Reviewer #1: Yes

Reviewer #2: Yes

5. Is the manuscript presented in an intelligible fashion and written in standard English?

Reviewer #1: Yes

Reviewer #2: Yes

6. Review Comments to the Author

Reviewer #1: The authors have adequately addressed the reviewers comments. The manuscript reads very well and has been strengthened by addressing the review.

Reviewer #2: In responding to my comments about including more detailed results in the abstract, the authors inserted the sentence "Primary care were concerned about the amount of time...". There is a missing word -- it should be primary care clinicians or primary care providers.

Otherwise all of my comments were adequately addressed.

7. PLOS authors have the option to publish the peer review history of their article (what does this mean?). If published, this will include your full peer review and any attached files.

Reviewer #1: No

Reviewer #2: No

---

## [Editor Report · Acceptance letter]

28 Feb 2020

PONE-D-19-33335R1 

Evaluating clinician acceptability of the prototype CanRisk tool for predicting risk of breast and ovarian cancer: a multi-methods study 

Dear Dr. Archer:

I am pleased to inform you that your manuscript has been deemed suitable for publication in PLOS ONE. Congratulations! Your manuscript is now with our production department. 

With kind regards,

on behalf of

Dr. Alvaro Galli 

Academic Editor

PLOS ONE